# Bioactive TNIIIA2 Sequence in Tenascin-C Is Responsible for Macrophage Foam Cell Transformation; Potential of FNIII14 Peptide Derived from Fibronectin in Suppression of Atherosclerotic Plaque Formation

**DOI:** 10.3390/ijms25031825

**Published:** 2024-02-02

**Authors:** Takuya Iyoda, Asayo Ohishi, Yunong Wang, Miyabi-Shara Yokoyama, Mika Kazama, Naoyuki Okita, Sachiye Inouye, Yoshimi Nakagawa, Hitoshi Shimano, Fumio Fukai

**Affiliations:** 1Department of Pharmacy, Faculty of Pharmaceutical Sciences, Sanyo-Onoda City University, Sanyo-Onoda 756-0884, Yamaguchi, Japan; 2Department of Molecular Patho-Physiology, Faculty of Pharmaceutical Sciences, Tokyo University of Science, Noda 278-8510, Chiba, Japan; 3Department of Endocrinology and Metabolism, Faculty of Medicine, University of Tsukuba, Tsukuba 305-8575, Ibaraki, Japan; 4Department of Complex Biosystem Research, Institute of Natural Medicine, University of Toyama, Toyama 930-0194, Toyama, Japan

**Keywords:** tenascin-C, atherosclerosis, macrophage, foam cell transformation, β1-integrin

## Abstract

One of the extracellular matrix proteins, tenascin-C (TN-C), is known to be upregulated in age-related inflammatory diseases such as cancer and cardiovascular diseases. Expression of this molecule is frequently detected, especially in the macrophage-rich areas of atherosclerotic lesions; however, the role of TN-C in mechanisms underlying the progression of atherosclerosis remains obscure. Previously, we found a hidden bioactive sequence termed TNIIIA2 in the TN-C molecule and reported that the exposure of this sequence would be carried out through limited digestion of TN-C by inflammatory proteases. Thus, we hypothesized that some pro-atherosclerotic phenotypes might be elicited from macrophages when they were stimulated by TNIIIA2. In this study, TNIIIA2 showed the ability to accelerate intracellular lipid accumulation in macrophages. In this experimental condition, an elevation of phagocytic activity was observed, accompanied by a decrease in the expression of transporters responsible for lipid efflux. All these observations were mediated through the induction of excessive β1-integrin activation, which is a characteristic property of the TNIIIA2 sequence. Finally, we demonstrated that the injection of a drug that targets TNIIIA2’s bioactivity could rescue mice from atherosclerotic plaque expansion. From these observations, it was shown that TN-C works as a pro-atherosclerotic molecule through an internal TNIIIA2 sequence. The possible advantages of clinical strategies targeting TNIIIA2 are also indicated.

## 1. Introduction

Atherosclerosis is defined as a chronic inflammatory disease of the arteries characterized by focal thickening of the inner portion of the artery wall, which can lead to a fatal myocardial or cerebral infarction. The first step of vascular inflammation would be initiated by immune cells accumulation through the events in cell adhesion called “rolling” and “firm adhesion”. These events are supported by various adhesion molecules, such as cell adhesion molecule (CAM), selectin, and integrin family members. Even in inflammatory atherosclerosis progression, increasing evidence has revealed the contribution of interactions between various CAMs expressed on endothelial cells and appropriate integrins on leukocytes, which play important roles in the abnormal accumulation of macrophages [1,2,3,4,5,6,7,8,9,10,11]. However, there are few reports describing in detail how intracellular signaling derived from integrin ligation leads to atherosclerotic plaque formation.

Integrins are heterodimeric molecules consisting of α and β chains and play pivotal roles during developmental and pathological processes. During these processes, integrins are known to work not only by anchoring cells to other cells or to the surrounding extracellular matrix (ECM), but also by activating intracellular signaling. Basically, integrin αβ heterodimers are expressed as their closed inactive forms, and they have no ability to interact with their ligands, CAMs, or ECM. Once cells are activated by hormonal factors, such as cytokines or chemokines, conformation of cell surface integrins will be changed to their open activated forms and become able to interact with the appropriate motifs in their ligands.

One of the integrin ligands, ECM, is an essential component of the microenvironment in the human body and works as a stimulator/regulator for the expression of various cellular functions. Therefore, construction of an appropriate ECM composition would be required for keeping the body healthy, and an altered composition in the ECM environment will often educe abnormal cellular function through the ECM–integrin signaling axis. During atherosclerosis progression, a remarkable decrease in the content of ECM molecules such as collagen and elastin fibers will frequently be observed, especially in the early stage. Additionally, accelerated expression of the proteases that work for ECM degradation has also been reported within the atherosclerotic aorta. Hence, it is presumable that the establishment of abnormal ECM composition could play a prominent role in the process of atherosclerosis progression.

At the site of inflammation, the induction of excessive tenascin-C (TN-C) expression is frequently observed [12,13,14,15,16,17]. TN-C is one of the ECM protein molecules and consists of a large number of variants because this molecule harbors several functional repeats, a part of which would be spliced alternatively during mRNA transcription. Among the TN-C variants, it has been reported that the molecules expressed in inflammatory conditions are large variants that harbor alternative splicing domains [17]. Therefore, investigations focusing on the role of alternative splicing domains in inflammation might be worthwhile for understanding the mechanisms underlying inflammatory disease progression.

Previously, we found a biologically active TNIIIA2 sequence from the FNIII-like repeat (FnIII)-A2 domain of TN-C [17], while this domain is known as one of the alternative splicing domains involved in TN-C variants overexpressed in inflammatory diseases. We reported that this sequence has the ability to promote aggressive cell adhesion through sustained and potentiated activation of α4/β1 and α5/β1 integrin by using a 22-mer peptide consisting of the TNIIIA2 sequence (Figure 1) [17]. It was shown that the TNIIIA2 peptide could bind to cell surface syndecan-4 (Sdc-4), and foamed TNIIIA2/Sdc-4 complexes will interact with the extracellular domain of β1-integrin molecules at the cell surface and make the conformation of β1-integrin active [17,18]. We also reported that the TNIIIA2 sequence is cryptic and exposed through limited digestion of parental TN-C by inflammatory proteases such as thrombin and matrix metalloproteases (MMPs) [17]. Afterwards, regarding age-related diseases, we demonstrated the potential of TNIIIA2 on the modulation of various cellular properties using malignant and/or premalignant tumor cells and fibroblasts [18,19,20,21,22].

Overexpression of TN-C has been detected during the progression of atherosclerosis, especially in macrophage-rich areas [12]. One of the hallmarks of atherosclerosis is accumulation of foam cells, which are macrophages with excessive cholesterol storage in their cytoplasm. Several groups have reported that TN-C plays a role in the regulation of macrophage functions; most of them used TN-C KO animals or recombinant TN-C without alternative splicing domains [23,24]. Considering the fact that the accelerated expression of MMPs would be widely detected in an inflammatory lesion, there is a possibility that the TNIIIA2 sequence would be exposed in the process of atherosclerosis progression. Thus, here, we tested the effect of the TNIIIA2 sequence on pro-atherosclerotic differentiation of macrophages and evaluated the clinical advantage of anti-atherosclerosis strategies targeting TNIIIA2.

## 2. Results

### 2.1. TNIIIA2 Stimulation in Intracellular Lipid Accumulation

Although transient upregulation of tenascin-C (TN-C) in atherosclerotic lesions has been reported, how TN-C contributes to atherosclerotic plaque progression in detail is still unknown. Progression of atherosclerotic plaque formation would be supported by a continuous supply of macrophages transformed into lipid-rich foam cells. Since we previously found a biologically active sequence termed TNIIIA2 in TN-C molecules, we first tested whether foam cell formation would be affected when macrophages were exposed to the TNIIIA2 sequence.

Mouse macrophage cell line RAW 264.7 cells were stimulated with a peptide containing the TNIIIA2 sequence (pTNIIIA2; Figure 1), and the intracellular lipid accumulation was evaluated by oil-red-O staining. As shown in the upper left panel of Figure 2A, untreated RAW 264.7 cells showed no lipid accumulation when they were co-cultured with 50 mg/mL of oxidized (ox)-LDL for 24 h. In contrast, extensive intracellular lipid accumulation was observed when RAW 264.7 cells were stimulated with 2.5 mg/mL of pTNIIIA2 and cultured in the presence of ox-LDL (upper right panel of Figure 2A, and black bar in Figure 2B). To exclude the possibility that the observed foam cell transformation induced by pTNIIIA2 is carried out by LPS contaminated in pTNIIIA2 solution, polymyxin-B, which could absorb LPS and inhibit LPS ligation to its receptor TLR4, was added simultaneously with pTNIIIA2. As shown in the lower right panel of Figure 2A, there is no difference in intracellular lipid accumulation between macrophages stimulated with pTNIIIA2 solely and pTNIIIA2/polymyxin-B. From these observations, the capability of the TNIIIA2 sequence to force the transformation of macrophages into atherosclerotic foam cells was indicated.

### 2.2. Molecular Mechanisms Underlying Macrophage Foam Cell Formation

To make clear the mechanism underlying TNIIIA2-induced acceleration in intracellular lipid accumulation (Figure 2), we next evaluated the phagocytic activity of macrophages exposed to the TNIIIA2 sequence. RAW 264.7 cells were stimulated with pTNIIIA2 for 3 h followed by co-culturing with carboxylate-modified latex beads (cell:beads = 1:20). Three field images randomly selected from each experimental group were taken using confocal microscopy, and then the phagocytic index (= “number of beads engulfed in cells” × “number of cells with engulfed beads”/“total cell number” in each field) was calculated as a parameter for phagocytic activity. Compared with the phagocytic activity shown by basal untreated macrophages, significant elevation was observed when macrophages were exposed to 2.5 mg/mL of pTNIIIA2 (Figure 3A, black bars). The elevated levels of phagocytic activity shown by TNIIIA2-stimulated macrophages were almost the same as those observed in cells treated with a typical inflammatory stimulus, LPS (Figure 3A, shaded bar). These results suggest that pTNIIIA2 might have the potential to accelerate phagocytic activity, as shown by its powerful inflammatory stimulus (Figure 3, black and shaded bar).

LPS-induced elevation in macrophage phagocytosis is known to be mediated through the MAPK signaling pathway [25,26]. Therefore, we then tested whether TNIIIA2 is capable of inducing activation of MAPKs. As shown in the upper panel of Figure 3B, pTNIIIA2 induced activation of p38 signaling in RAW 264.7 cells, whereas there was no difference in the ERK phosphorylation status. In contrast, as described in previous reports [26], activation of both ERK and p38 signaling was detected in LPS-stimulated macrophages (Figure 3C). Therefore, we next tried to evaluate the effect of p38 inhibition on the phagocytic activity of macrophages exposed to pTNIIIA2. As shown in Figure 3A, enhanced phagocytic activity induced by pTNIIIA2 (black bar) was completely abolished when cells were pretreated with the p38 inhibitor SB203580 (hatched bar in center). In sharp contrast, elevated phagocytosis induced by LPS was not affected by SB203580 (Figure 3A, shaded and hatched bar in right). These observations suggest that the mechanism underlying acceleration of phagocytosis induced by TNIIIA2 would be different from that observed in the inflammatory response.

To clarify the molecules responsible for enhanced phagocytosis observed in pTNIIIA2-treated macrophages, the mRNA expression of the molecules relating to the recognition of the phagocytic target was evaluated. Among the phagocytic receptors tested, LOX1, a lectin-type ox-LDL receptor 1, was significantly elevated when macrophages were stimulated with pTNIIIA2 (Figure 3D, black bar), while TNIIIA2-induced overexpression of LOX1 was completely abolished by SB203580 pretreatment (Figure 3C, hatched bar). On the other hand, as for the other receptors, no significant differences in SR-A and CD36 expression were detected in our experimental conditions (Figure 3E). From these observations, foam cell transformation observed in TNIIIA2-treated macrophages might be promoted through p38-dependent upregulation of LOX1 and following LOX1-mediated recognition of ox-LDL, at least in part.

### 2.3. Effect of pTNIIIA2 on Lipid Transporter Expression

Although TNIIIA2 showed the ability to enhance phagocytosis significantly, it was just ~30% elevated when compared to untreated macrophages (Figure 3A). Considering how much lipid droplet accumulation was promoted in pTNIIIA2-treated RAW 264.7 cells (Figure 2), it seemed to not be enough to explain the underlying mechanism of the TNIIIA2-mediated increase in foam cell transformation only by acceleration of phagocytosis. Thus, next, we focused on molecules responsible for cholesterol efflux. Physiologically, macrophages engulf the surplus supply of oxidized LDL molecules and put them back into circulation in HDL form. In this cholesterol transfer process, ATP-cassette transporter family molecules ABCA1 and ABCG1 are known to play a role in the latter lipid efflux event. As shown in Figure 4A, mRNA expression of ABCA1 and ABCG1 was significantly decreased in macrophages stimulated with pTNIIIA2. Suppressed expression of ABCA1 and ABCG1 in TNIIIA2-stimulated macrophages was also confirmed at the protein level by using Western blotting (Figure 4B). The effect of pTNIIIA2 on ABCA1 and ABCG1 expression was completely attenuated by inhibition of p38 signaling (Figure 4A). These results suggest that the cholesterol efflux in macrophages exposed to TNIIIA2 sequences would be suppressed through downregulation of outward lipid transporters in a p38-dependent manner.

### 2.4. β1-Integrin Activation in TNIIIA2-Mediated Macrophage Foam Cell Transformation

As mentioned above, TNIIIA2 was found to be a bioactive sequence capable of inducing β1-integrin activation. Therefore, we next investigated whether the foam cell transformation observed in TNIIIA2-stimulated macrophages shows dependence on β1-integrin activation status. Previously, we reported that the TNIIIA2-mediated activation of β1-integins would not be promoted through general “inside-out” signaling but would be achieved by an “outside-out” one. TN-C fragments containing the TNIIIA2 sequence would bind to cell surface syndecan-4 (Sdc4) first, and then TNIIIA2/Sdc4 complexes would interact laterally with β1-integrins, followed by making the conformation of integrins active [17]. This mechanism might be important for making TNIIIA2-mediated β1-integrin activation more sustained and forceful than that induced by general integrin activators such as cytokines/chemokines. Based on these facts, we next employed 9EG7, an antibody inducing β1-integrin activation through the “outside-out” fashion, and then evaluated the effect of this antibody on macrophage foam cell transformation. When the expression of the phagocytic receptor was tested, 9EG7 addition showed significant upregulation of LOX1 (Figure 5A, white and shaded bar), although the level of increase was less than that observed in pTNIIIA2-stimulated macrophages (Figure 5A, black and shaded bar). The expression of lipid transporters ABCA1 and ABCG1 was also suppressed in macrophages stimulated by 9EG7 (Figure 5B, white and shaded bar), almost the same as that in pTNIIIA2-stuimulated macrophages (Figure 5B, black and shaded bar).

On the other hand, in sharp contrast to the TNIIIA2 sequence, we have reported the other biologically active amino-acid sequence derived from the fibronectin molecule and termed the peptide including this sequence as “FNIII14” [27,28]. FNIII14 would bind to and form a complex with the cell surface eEF1A molecule. Formed FNIII14-eEF1A complexes could interact with the extracellular domain of β1-integrin molecules at the cell surface and make the conformation of β1-integrin inactive, potently. To evaluate not only the β1-integrin dependency of TNIIIA2-mediated foam cell transformation, but also the potential of FNIII14 as an anti-atherosclerotic drug candidate, we next tested whether FNIII14 addition could rescue macrophages from pTNIIIA2-mediated foam cell transformation. As shown in Figure 5A, LOX1 expression suppressed by pTNIIIA2 addition was completely attenuated by simultaneous addition of FNIII14 (Figure 6A, left panel, hatched bar). The pTNIIIA2-mediated down-regulation in ABCA1 and ABCG1 was also completely canceled when FNIII14 was added to the medium used for pTNIIIA2 stimulation (Figure 6A, center and right panel, hatched bar). Consistent with the expression of these receptors and transporters, both the phagocytic activity and intracellular lipid content that were accelerated in pTNIIIA2-stimulate macrophages were significantly suppressed by additional FNIII14 stimulation (Figure 6B, hatched bar, and Figure 6C). Taking all the results shown in Figure 5 and Figure 6 into consideration, TNIIIA2-promoted foam cell transformation would be mediated through induction of β1-integrin activation and following intracellular signaling activation.

### 2.5. Forced Inactivation of β1-Integrin in Atherosclerosis Model Mouse

Since the β1-integrin inactivator FNIII14 showed the ability to significantly suppress pTNIIIA2-induced macrophage foam cell transformation, we finally tested the potential of FNIII14 as a therapeutic anti-atherosclerotic drug. Male LDLR^−/−^ mice were fed normal chow until 10 weeks of age and then were changed to a Western diet. At 12 weeks of age, the mice were randomly divided into 3 groups: Group #1: vehicle control injection, Group #2: control peptide injection, and Group #3: FNIII14 peptide injection, and were started on the administration of each peptide or vehicle every other day intravenously at 100 mg/head (Figure 7A). Then, at 21 weeks of age, these mice were sacrificed, and their thoracic–abdominal aortas were collected. Atherosclerotic legions in collected aortas were visualized by Sudan red staining, and the progression of atherosclerosis was evaluated through the percentage of Sudan red positive area vs. total aortic area. As shown in Figure 7B, the atherosclerotic plaque area was significantly suppressed in mice with FNIII14 injections. Compared with vehicle-injected mice, FNIII14-injected animals showed an approximately 40% decrease in the size of atherosclerotic lesions (Figure 7C). Since injection of the control peptide showed no effect on the pathological condition, the observed suppression in atherosclerotic plaque formation would be promoted through expression of FNIII14’s bioactivity. Of note, there were no differences in body weight between the groups. These results suggest that atherosclerotic plaque formation in vivo would be promoted through excessive activation of β1-integrin, possibly induced by the TNIIIA2 sequence, and would be attenuated by the administration of drugs targeting the bioactivity of TNIIIA2.

## 3. Discussion

It is well known that the atherosclerotic lesion shows a high level of TN-C expression. However, the pathological role of TN-C in atherosclerotic plaque formation has not been unveiled. Here, we showed that the 22-mer peptide consists of a TNIIIA2 sequence, which is present in the FNIII-like repeat (FnIII)-A2 domain of TN-C and has the ability to transform macrophages into pro-atherosclerotic foam cells. TNIIIA2-mediated foam cell transformation was carried out through β1-integrin activation, and the atherosclerotic plaque formation observed in atherosclerosis model animals was significantly suppressed by injection of a reagent promoting β1-integrin inactivation. These results suggest that de-regulation of β1-integrin activity and the following intracellular signaling activation would have the potential to become a fruitful clinical target against the progression of atherosclerosis.

In this study, we focused on the ECM protein TN-C, the expression of which was often elevated specifically at the inflammatory site. Since TN-C expression would rarely be detected in a healthy adult body [12,13], TN-C has attracted attention as a therapeutic target for clinical treatment against various inflammatory diseases, including atherosclerosis. However, until now, we have not succeeded in establishing any clinical strategies/drugs targeting TN-C for managing inflammatory disease status. The clinical values of TN-C are accomplishing nothing more than using this molecule as a biomarker for inflammatory disease progression. One of the reasons for the difficulties in establishing clinical strategies targeting TN-C could be the complexity in the process of TN-C’s function. TN-C consists of various functional modular structures such as “hapted repeats”, “EGF-like repeats”, “FnIII-like repeats”, and “fibrinogen-like globe (FBG)”. Among these modules, FnIII-like repeats could be further divided into two groups, constitutively including FnIII-1 to -8 domains and the alternatively spliced FnIII-A1 to -A4, -AD2, -AD1, -C, and -D domains (see Figure 1). Therefore, the TN-C molecule has a large number of variants. Although the molecular functions of TN-C would be determined by the combination of functional modules exposed on the molecular surface, structurally, it has also been well known that TN-C molecules would be processed post-transcriptionally by various proteases, accompanied by changes in their conformation. Because these convoluted processes are responsible for determining the expression of their molecular functions, the role of TN-C in disease progression is still controversial in terms of whether this molecule is protective or harmful.

Even in atherosclerosis, the role of TN-C in disease progression is still widely discussed, whereas elevated expression of TN-C harboring alternative splicing domains is well described [12,13]. With regard to its anti-atherosclerotic protective effects, Wang et al. have reported that TN-C protects against the development of atherosclerotic lesions through suppressed expression of eotaxin, an inflammatory CC chemokine [24]. They have also shown that TN-C could suppress intraplaque hemorrhages observed in atherosclerosis [29]. Consistent with these reports, the protective role of TN-C in other blood vessel diseases, such as aortic dissection, has been reported [30]. In sharp contrast are the harmful effects of TN-C on vascular diseases. Liu et al. have shown that TN-C could accelerate macrophage foam cell transformation through TN-C binding to the LPS receptor TLR4 [31]. Disease-promoting effects of TN-C on artery graft stenosis and neointimal hyperplasia have also been reported [32,33].

To solve the inconsistency of the role of TN-C in the inflammatory disease progression described above, investigations focused on each singular functional motif/module included in TN-C would be beneficial. Actually, Piccinini et al. have demonstrated that the ligation of the FBG domain to Toll-like receptor 4 (TLR4) is responsible for the inflammatory polarization of macrophages induced by TN-C [34]. Luo et al. have also reported that the inhibition of TN-C-induced foam cell formation would be carried out by ATF3-mediated downregulation of TLR4 [35]. In this study, as a new insight in the investigations focused on the role of each bioactive region in TN-C during atherosclerosis progression, we demonstrated here that the TNIIIA2 sequence in TN-C has the ability to accelerate macrophage transformation into foam cells. TNIIIA2-induced intracellular lipid accumulation seemed to be mediated through both increased LDL uptake and decreased HDL efflux. TNIIIA2-induced foam cell formation was mediated through β1-integrin activation. The reagent making β1-integrin inactive, FNIII14, showed significant inhibition in atherosclerotic plaque formation in an in vivo mouse model, as well as macrophage foam cell transformation in vitro. Although there is a possibility that the macrophage foam cell formation induced by FBG/TLR4 ligation also occurs through β1-integrin activation, the TNIIIA2 sequence, which is included in the inflammatory site-specific TN-C, would work as a pro-atherosclerotic molecule. Thus, the TNIIIA2-sequence and its bioactivity might be a potential clinical target.

Atherosclerotic plaques would be ruptured at the end stage, and this event would become a cause of fatal cardiac and cerebral infarction. Therefore, keeping atherosclerotic plaques stable would be clinically required. It has been well accepted that the instability of plaques has a positive correlation with foam cell content as well as the normalcy of vascular endothelial wall status. Regarding the risk factor of the former, we showed the possibility here that attenuation of TNIIIA2′s bioactivity would contribute to increased plaque stability because it could suppress macrophage foam cell transformation. On the other hand, for the latter factor, accumulating evidence has demonstrated that HDL cholesterol has the potential to protect endothelial cells in various vascular diseases, including atherosclerosis [36,37,38,39]. HDL maturation would be carried out by receiving cholesterol exported from surrounding cells, including macrophages, through ABCA1 and ABCG1 expressed on the cell surface. Since we showed here that the expression of these transporters in macrophages was suppressed by TNIIIA2 stimuli (see Figure 4), it would be presumed that TNIIIA2 might act to decrease the content of mature HDL in circulation. If so, FNIII14-mediated neutralization of TNIIIA2′s bioactivity would contribute to decreasing the risk of plaque rapture by both lowering the foam cell content in atherosclerotic plaques and keeping the endothelial cell wall healthy. Because abnormalities in endothelial cells derived from TNC-KO have also been reported [29], investigations about the effect of TNIIIA2 and FNIII14 sequences on endothelial cells will be part of our next targets.

As described above, macrophages play a pivotal role in the regulation of atherosclerosis development. It has been well accepted that the majority of these macrophages infiltrate from the bloodstream into the sub-endothelial space of the vessel through cell adhesion-molecule-dependent events termed “rolling” and “firm adhesion”. The mechanisms of these events have been well documented, such that the former rolling step is mediated through interactions of cell-surface glycoproteins with L- and E-selectin, and the latter “firm adhesion” is brought about by the interactions between ICAM-1 with α_L_β2 integrin (LFA1) and/or VCAM-1 with α4β1 integrin (VLA4). Even in atherosclerosis, the contribution of VCAM-1 to monocyte accumulation in the early stages of disease progression has been reported [8,9]. Therefore, there is a possibility that the observed amelioration of atherosclerotic plaque formation, as promoted by injection of the β1-integrin inactivator FNIII14 peptide (see Figure 7 of this study), was just a reflection of the decrease in the number of infiltrated macrophages. However, by contrast, it has also been reported that anti-VCAM-1 antibody-mediated suppression of atherosclerotic plaque formation was partial, whereas these antibodies were powerful inhibitors of U937 monocytic cell adhesion on HUVEC [40]. Additionally, O’Brien et al. reported that the majority of VCAM-1-positive cells in atherosclerotic plaques were not endothelial cells but smooth muscle cells and macrophages [41]. Moreover, several groups reported that the contribution of VLA-4 on monocyte/macrophage infiltration into the sub-endothelial space of blood vessels might be partial [1,2,42] and that ICAM-1/β2-integrin interaction plays a key role in macrophage accumulation in inflammatory conditions, including atherosclerosis [2,3,4,5,6,7]. Consistent with these reports, Nakashima et al. reported that the elevated expression of VCAM-1 observed in the atherosclerosis model animal was limited in the aortic sinus and a small area of outflow at branch sites [43]. Since the atherosclerotic plaque would be developed mainly at the sites exposed to turbulent blood-flow-mediated shear stress, such as the aortic arch, descending aorta, and abdominal aorta, the contribution of VCAM-1 and its receptor α4β1-integrin to macrophage infiltration steps seemed to be relatively low. This estimation would be supported by the fact that the VCAM-1 promoter does not have shear-stress response elements (SSREs), whereas the ICAM-1 promotor has [44]. There is another report showing that the adhesion of human monocytic THP-1 cells to endothelial cells was not inhibited by siRNA-mediated knockdown of α4β1, αLβ2, and αMβ2 on THP-1 cells [45]. Taking all observations above into consideration, the anti-atherosclerotic effect observed in FNIII14 injection would be promoted not only through suppression of macrophage infiltration but also by inhibition of foam cell transformation.

## 4. Materials and Methods

### 4.1. Reagents

Peptide TNIIIA2, peptide FNIII14, and its analogous inactive scrambled control peptide FNIII14scr have been previously described. Anti-ABCA1 antibody was purchased from Millipore. Anti-ABCG1 antibody was obtained from SantaCruz Biotech Inc. (Dallas, TX, USA). Anti-phospho-ERK antibody, anti-phospho-p38 antibody, and p38 inhibitor SB203580 were obtained from Cell Signaling Tech (Danvers, MA, USA). Anti-β1-integrin antibody (ab179471) was obtained from abcam PLC (Cambridge, UK). β1-integrin activation antibody 9EG7 and its isotype control antibody were obtained from BD Biosciences (Franklin Lakes, NJ, USA).

### 4.2. Cell Culture and ox-LDL Loading

Mouse macrophage cell line RAW 264.7 cells were maintained in low-glucose DMEM containing 10% heat-inactivated FBS and penicillin/streptomycin. This cell was seeded at 12.5 × 10^4^ cells/well on a 24-well plate and cultured for 2.5 h, followed by changing medium containing 1% FBS. Two and a half hours later, human oxidized LDL (Biomedical Tech Inc., Madrid, Spain) was added to the culture at 50 mg/well simultaneously with 2.5 mg/mL of pTNIIIA2, followed by a 24 h incubation at 37 °C, 5% CO_2_. Then, these cells were washed with PBS, fixed with 10% neutral buffered formalin, and applied to oil-red-O staining.

### 4.3. Phagocytic Assay

RAW 264.7 cells were seeded on a glass-bottom plate at 25 × 10^4^ cells/mL and cultured for 3 h. Then, the culture medium was changed to DMEM containing 1% FBS with/without 2.5 mg/mL TNIIIA2. Two and a half hours after TNIIIA2 stimulation, carboxylate-modified latex beads (Molecular Probes, FluoSpheres, 2 mm*ϕ*) were added to the cell culture at the ratio of cells:beads = 1:20, and were co-cultured with macrophages for 20 min at 37 °C. Then, unbound or unengulfed beads were washed out with PBS and continuously cultured for 15 min more in phenol-red-free DMEM for the completion of bounded bead engulfment. The number of engulfed beads (#1), bead-engulfed macrophages (#2), and total cells (#3) in three random confocal microscopic views was calculated and compared to the phagocytic index (=#1 × #2/#3) of each group.

### 4.4. Polymerase Chain Reaction Assay

The mRNA expression was evaluated using real-time PCR with the SYBR-green system. Total RNA was isolated from RAW 264.7 cells using an RNeasy mini-prep kit (Qiagen Inc., Hilden, Germany), and cDNA was prepared from the isolated total RNA using a QuantiTect reverse transcription kit (Qiagen). Relative mRNA expression was evaluated using Thunderbird SYBR qPCR Mix (Toyobo, Osaka, Japan). Employed primer sets, as shown in Table 1, were synthesized referencing published papers [46,47,48].

### 4.5. Animal Study

LDLR^−/−^ mice were purchased from the Jackson Laboratory (Bar Harbor, ME) and maintained on a normal diet (Oriental Yeast Company, Tokyo, Japan) with a 14 h light/10 h dark cycle. For induction of atherosclerosis, at 10 weeks of age, the mice were fed WD (D12079B [34% sucrose, 21% fat, 0.15% cholesterol]; Research Diets, Inc., New Brunswick, NJ, USA) under the indicated conditions. At 12 weeks of age, the mice were randomly divided into three groups, followed by intravenous injection of pFNIII14, pFNIII14scr, or saline, respectively, every other day. At 21 weeks of age, the mice were euthanized to extract their aortas, and the aortas were cut along the midline from the iliac arteries to the aortic root, pinned flat, and stained with Sudan IV. Sudan IV-positive atherosclerotic lesions were quantified using Photoshop software version 24.4.1 (Adobe Systems Inc., San Jose, CA, USA). All animal husbandry procedures and animal experiments were performed in accordance with the Regulation of Animal Experiments of the University of Tsukuba and were approved by the Animal Experiment Committee, University of Tsukuba (#19-270, 2019/6/1).

### 4.6. Statistical Analysis

Results are expressed as means ± standard deviation. Differences between experimental groups were analyzed using one-way factorial ANOVA and a post-hoc test. When a *p*-value was less than 0.05, the difference was considered statistically significant.

## 5. Conclusions

We demonstrated in this thesis that TN-C has the potential to act as a pro-atherosclerotic through its internal TNIIIA2 sequence. This sequence is capable of accelerating macrophage–foam cell transformation. Clinical values of strategies targeting TNIIIA2 or TNIIIA2-inducible excessive β1-integrin activation in atherosclerosis treatment were shown. FNIII14 peptide would be a good candidate for a new anti-atherosclerosis drug, at least for decreasing the number of foam cells in plaques, although further studies would be needed.

## Figures and Tables

**Figure 1 ijms-25-01825-f001:**
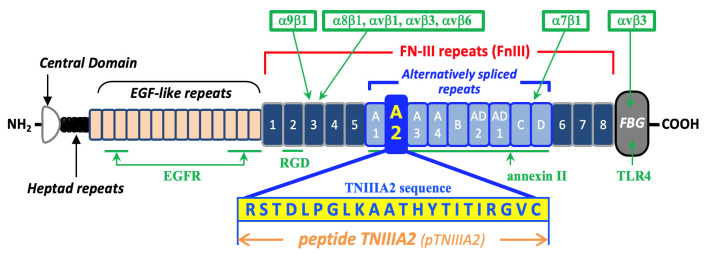
Schematic illustration of the structure of tenascin-C molecule and the distribution of bioactive regions/sequences. The sites responsible for receptor binding are shown with a green arrow with the names of corresponding interaction molecule/s. This figure was modified from a published article written by our group [18].

**Figure 2 ijms-25-01825-f002:**
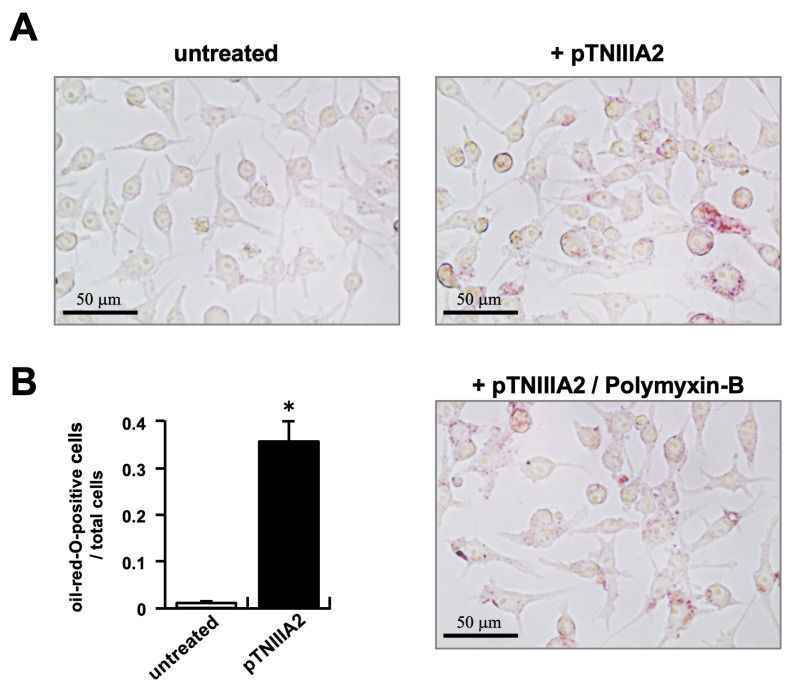
Accelerated accumulation of intracellular lipids in RAW 264.7 macrophages exposed to TNIIIA2, cultured with ox-LDL for 24 h. (**A**) Accumulated intracellular lipid droplet was stained with oil-red-O. Typical image is shown. (**B**) Number of oil-red-O positive cells in each field was counted and the ratio of it against total cell number was calculated. Three field images randomly selected from each experimental group were subjected to quantification. *; *p* < 0.05 vs. inhibitor-free untreated control group (N = 3, n = 9).

**Figure 3 ijms-25-01825-f003:**
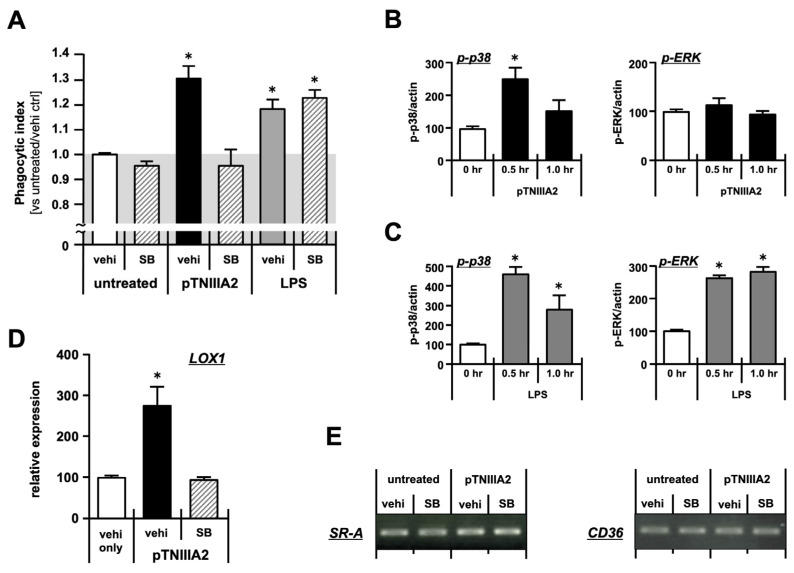
pTNIIIA2 stimulates phagocytosis through p38 signaling. (**A**) Macrophage phagocytic activity against carboxylate-modified microbeads was evaluated using confocal microscopy. Calculated phagocytic index is shown in bar graph (N = 3, n = 9). (**B**,**C**) Activation status of p38 and ERK in RAW 264.7 cells stimulated with pTNIIIA2 (**B**) or LPS (**C**) was quantified from western blotting image using image J software version 1.53 (n = 3). (**D**) Real-time PCR analysis of the gene expression of phagocytic LOX-1 receptor in pTNIIIA2-treated RAW 264.7 cells with/without p38 inhibitor, SB (n = 3). (**E**) RT-PCR analysis of the gene expression of SR-A and CD36 in pTNIIIA2-treated RAW 264.7 cells (n = 3). *; *p* < 0.05 vs. SB-free untreated control group. vehi: vehicle for SB203580, SB: SB203580.

**Figure 4 ijms-25-01825-f004:**
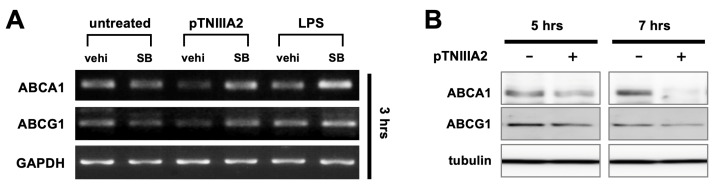
Effect of pTNIIIA2 stimulation on the expression of cell surface transporters responsible for lipid efflux. Expression of ABCA1 and ABCG1 in RAW 264.7 macrophages was evaluated in both mRNA level and protein level (**A**,**B**) (n = 3). vehi: vehicle for SB203580, SB: SB203580.

**Figure 5 ijms-25-01825-f005:**
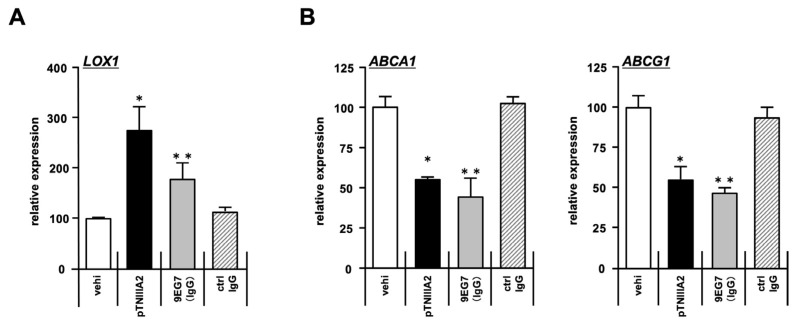
Alteration in LOX1, ABCA1, and ABCG1 expression was also induced by the other β1-integrin-activation reagent. (**A**) LOX1 mRNA expression was increased when RAW 264.7 macrophages were stimulated by β1-integrin activation antibody 9EG7, as well as by pTNIIIA2. (**B**) ABCA1 and ABCG1 mRNA expression was decreased by β1-integrin activation antibody 9EG7, as well as by pTNIIIA2. *; *p* < 0.05 vs. SB-free untreated control group. **; *p* < 0.05 vs. control IgG-treated group (n = 3).

**Figure 6 ijms-25-01825-f006:**
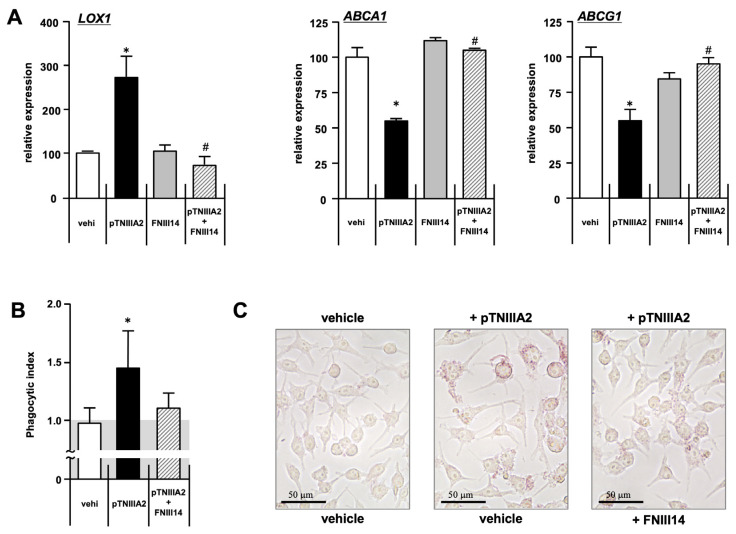
β1-integrin inactivation would keep RAW 264.7 cells away from TNIIIA2-mediated elicitation of pro-atherosclerotic cellular functions. (**A**) pTNIIIA2-mediated increase in LOX1 expression (left panel) and decrease in ABCA1/G1 expression (middle and right panels) were almost completely canceled in macrophages pretreated with β1-integrin inactivator FNIII14 (n = 3). (**B**) pTNIIIA2-induced elevation in phagocytic activity was also neutralized by FNIII14 (n = 3). (**C**) In the end, FNIII14-treated macrophages would be rescued from excess accumulation of intracellular lipid droplets induced by pTNIIIA2 (N = 3, n = 9). *; *p* < 0.05 vs. untreated control group. #; *p* < 0.05 vs. TNIIIA2 solely administrated group.

**Figure 7 ijms-25-01825-f007:**
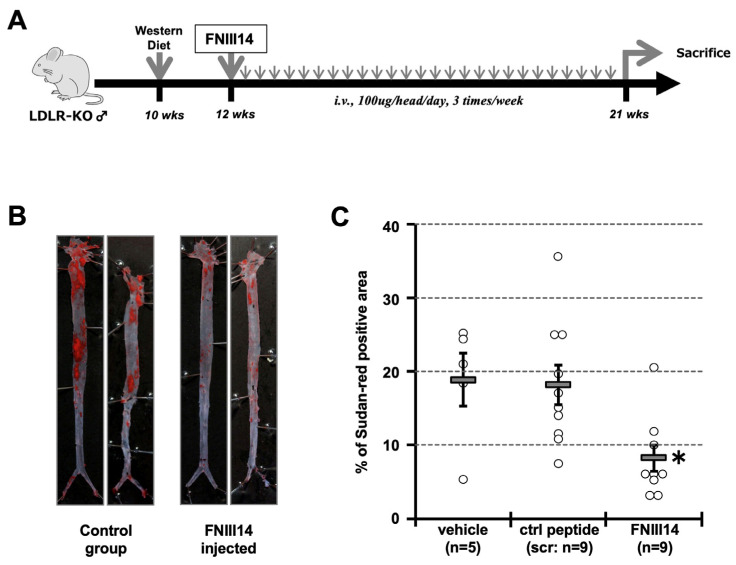
Values of peptide-mediate β1-integrin inactivation in atherosclerotic plaque formation using LDLR^−/−^ mice. (**A**) schematic illustration of in vivo examination schedule. (**B**) Typical images of Sudan-red-stained aorta, which were isolated from control peptide-injected mice and FNIII14 injected mice, are shown. (**C**) The percentage of Sudan-red-positive area in each aorta was calculated and compared. *; *p* < 0.05 vs. control peptide-injected group.

**Table 1 ijms-25-01825-t001:** Sequence of primers used in this study.

Target Gene	Primer Sequence			
LOX1	Fw:	5′-	GTGGACACAATTACGCCAGGTA	-3′
	Rv:	5′-	GCCCTTCCAGGATACGATCC	-3′
ABCA1	Fw:	5′-	CTTCCCACATTTTTGCCTGG	-3′
	Rv:	5′-	AAGGTTCCGTCCTACCAAGTC	-3′
ABCG1	Fw:	5′-	CTGTCTGATG GCCGCTTTCT	-3′
	Rv:	5′-	CTGGACACGACCTCGTCCAC	-3′
SR-A	Fw:	5′-	CATGAACGAGAGGATGCTGACT	-3′
	Rv:	5′-	GGAAGGGATGCTGTCATTGAA	-3′
CD36	Fw: Rv:	5′- 5’-	TCCAGCCAATGCCTTTGC TGGAGATTACTTTTCAGTGCAGAA	-3′ -3’
GAPDH	Fw:	5′-	TTCACCACCATGGAGAAGGC	-3′
	Rv:	5′-	GGCATGGACTGTGGTCATGA	-3′

## Data Availability

The data presented in this study are available, upon request, from the corresponding author.

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
