# Peer review of "Bioactive TNIIIA2 Sequence in Tenascin-C Is Responsible for Macrophage Foam Cell Transformation; Potential of FNIII14 Peptide Derived from Fibronectin in Suppression of Atherosclerotic Plaque Formation"

_ijms, 2024, doi:10.3390/ijms25031825_

Round 1

Reviewer 1 Report

Comments and Suggestions for Authors

In this manuscript (ID# ijms-2813280), entitled “Bioactive TNIIIA2 Sequence in Tenascin-C is Responsible for Macrophage Foam Cell Transformation; Potential of Anti-adhesive Peptide FNIII14 in Suppression of Atherosclerotic plaque formation”, authors, Iyoda et al studied the effect of TNIIIA2 and FNIII14 peptides in atherosclerosis and macrophagic activity using macrophage cell-line and a high fat diet-treated mouse model. Their results indicate that TNIIIA2 increases intracellular lipid accumulation in microphages via a p38-dependent signaling pathway and that FNIII14 attenuates atherosclerotic plague formation in high fat diet-treated mice. The results are interesting. However, the experimental design is not rigorous and some results are not reliable. Several major concerns have been listed in the following paragraphs:

1. In Figure 4, the expression of ABCA and ABCG was studied using RT-PCR. The mRNA levels should be confirmed using real-time PCR, which more accurate. In addition, the cholesterol is collected into HDL on endothelial cells. What is the role of ABCA and ABCG in microphages?

2. What is the effect of TNIIIA2 on atherosclerotic plaque formation as compared with FNIII4 in this mouse model? What is the effect of TNIIIA2 plus FNIII14? The most important piece of data is missing.

3. The pharmacokinetic information of FNIIIA2 is unclear. After intravenous administration, how FNIIIA2 penetrated into vascular wall, acting on macrophages in vascular wall?

4. What is the effect of FNIIIA2 and TNIIIA2 on beta-integrin activity?        

5. Please provide a clear conclusion from this study. 

6. In page 5, Lines 173-174, authors described “no significant differences in SR-A, CD36, and LDLR expression was detected in our experimental conditions”. I could not find this result.

Comments on the Quality of English Language

The English language is find. only minor editing of English language is required.

Author Response

We appreciate your quite productive and insightful suggestions in refereeing. Our point-by-point actions to each of the comments are in attachment pdf file. So, see the uploaded file please. Thank you for your co-operation.

Reviewer 2 Report

Comments and Suggestions for Authors

The authors present an interesting study examining the influence of a particular sequence of the tenascin-c extracellular matrix protein; namely TNIIIA2, on macrophage behaviour relative to atherosclerotic plaque development. Briefly, the authors exposed macrophage cultures to the TNIIIA2 sequence, and measured a number of indices relevant to a pro-atherosclerotic phenotype; phagocytic activity and lipid efflux transporters,  with treatment found to promote uptake of lipid molecules by the cells. Finally, animal studies demonstrated that amelioration of TNIIIA2’s activity reduced plaque progression, corroborating the in vitro work described earlier in the article. Taken together, TNIIIA22 represents a potentially viable therapeutic target for future studies.

In reviewing the manuscript I made a number of observations. The following should be considered by the authors when preparing a suitable revision.

1.       For the macrophage activation was there any measurement of the total levels for ERK and p38 to determine the whole activation state of the protein?

2.       The n-numbers need to be clearly stated in each figure legend.

3.       Why is there a difference in n-number in Figure 7 (animal study)?

4.       What was the source of the primer sequences and were they found to be MIQE guidelines compliant?

5.       At times in the text there is reference in the figures to ‘hatched lines’ or patterns, but this not seem to be apparent in the figures reviewed.

6.       For the PCR expression data where agarose gels are presented why could quantitative data via PCR not be provided for this?

7.       In those figures where protein expression is being presented, western blots and quantitative data should be presented at all times. There are instances where one or the other is presented, but rarely both.  

Author Response

We appreciate your quite productive and insightful suggestions in refereeing. Our point-by-point actions to each of the comments are in attached file. So, please see the uploaded pdf. Thank you for your co-operation.

Reviewer 3 Report

Comments and Suggestions for Authors

The importance of macrophages in the pathophysiology of atherosclerosis cannot be overstated, prompting researchers to look for strategies to alter the severity of the disease by modifying macrophage functional potential.

The scientists developed a novel approach to assess the impact of alternative splicing locations on the functional status of the rat macrophage lineage in vitro.

Given that the expression level of this molecule increases in several pathological processes, including vascular disorders, the study's relevance is undeniable. The findings from the study on the role of a specific tenascin-c locus in modifying macrophage functional status and the mechanisms underlying this phenomenon undoubtedly expand fundamental ideas about the role of this molecule in pathological processes, particularly in the pathogenesis of atherosclerosis.

The scientists noticed that TNIIIA2 plays a role in macrophage acquisition of foam cell shape via effects on p38, LOX1, modulation of ABCA1 and ABCG1 efflux pump activity, and activation of beta-integrins. These are mostly new and intriguing fundamental findings.

The results are indisputable because they are based on data acquired during the study process. One limitation of the study is that the authors primarily refer to works older than 5 years.

As part of the conversation, I'd need clarification or the writers' perspectives on the following points:

1) Based on a vival study, the scientists propose that FNIII14 could be employed in the future as a chemical capable of suppressing excessive fat accumulation in macrophages.  TNC-/- and /apo E-/- knockout animals, on the other hand, have enhanced macrophage adherence to endotheliocytes (Wang L, Wang W, Shah PK, Song L, Yang M, Sharifi BG. Tenascin-C gene deletion worsens atherosclerosis and causes intraplaque bleeding in Apo-E-deficient animals. doi: 10.1016/j.carpath.2011.12.005.) The point is, have you noticed similar alterations in animals, or was this not one of the study objectives?

2) TLR4 and CD36 have been implicated in foam cell formation (Liu R, He Y, Li B, Liu J, Ren Y, Han W, Wang X, Zhang L. Tenascin-C produced by oxidized LDL-stimulated macrophages increases foam cell formation via Toll-like receptor-4. Mol Cells. 2012 Jul;34(1):35-41. doi: 10.1007/s10059-012-0054-x. TLR4 and siCD36 antibodies inhibit tenascin's action. Have you noticed something similar?

(3) One of the variables involved in regulating the complex process of foam cell production is activating transcription factor 3 (ATF3). Furthermore, by decreasing TLR-4 in LPS-stimulated THP-1 macrophages, ATF3 can operate as a negative regulatory factor to limit TN-C-induced foam cell formation (Luo H, Wang J, Qiao C, Zhang X, Zhang W, Ma N. ATF3 Suppresses TLR-4 and Inhibits Tenascin-C-Induced Foam Cell Formation in LPS-Stimulated THP-1 Macrophages. J Atheroscler Thromb. 2015;22(11):1214-23. doi: 10.5551/jat.28415.). Can the chemical you obtained have the same effects?

4) Polymyxin B, an antibiotic that may generate holes and is used to transfect macrophages with viruses (Jones CH, Rane S, Patt E, Ravikrishnan A, Chen CK, Cheng C, Pfeifer BA. Treatment with polymyxin B improves bactofection efficacy while decreasing cytotoxicity. (Fathalla AM, Chow SH, Naderer T, Zhou QT, Velkov T, Azad MAK, Li J. Mol Pharm. 2013 Nov 4;10(11):4301-8. doi: 10.1021/mp4003927.) and induces cell apoptosis (Fathalla AM, Chow SH, Naderer T, Zhou QT, Velkov T, Azad MAK, Li J. Polymyxin-Induced Cell Death in Human Macrophage-Like THP-1 and Neutrophil-Like HL-60 Cells Is Associated with Apoptotic Pathway Activation. Antibacterial Agents 2020 Aug 20;64(9):e00013-20. doi: 10.1128/AAC.00013-20. What was the purpose of introducing it to macrophages?

Author Response

(The authors gave the same response as above.)

Round 2

Reviewer 1 Report

Comments and Suggestions for Authors

no further recommendation

Reviewer 2 Report

Comments and Suggestions for Authors

The authors have suitably addressed my comments